# RAS Subcellular Localization Inversely Regulates Thyroid Tumor Growth and Dissemination

**DOI:** 10.3390/cancers12092588

**Published:** 2020-09-10

**Authors:** Yaiza García-Ibáñez, Garcilaso Riesco-Eizaguirre, Pilar Santisteban, Berta Casar, Piero Crespo

**Affiliations:** 1Instituto de Biomedicina y Biotecnología de Cantabria (IBBTEC), Consejo Superior de Investigaciones Científicas (CSIC)-Universidad de Cantabria. Santander, E-39011 Cantabria, Spain; yaiza5991@gmail.com (Y.G.-I.); berta.casar@unican.es (B.C.); 2Instituto de Investigaciones Biomédicas “Alberto Sols”, Consejo Superior de Investigaciones Científicas -Universidad Autónoma de Madrid. E-28029 Madrid, Spain; griesco@iib.uam.es (G.R.-E.); psantisteban@iib.uam.es (P.S.); 3Departamento de Endocrinología y Nutrición, Hospital Universitario de Móstoles, E-28935 Madrid, Spain; 4Departamento de Endocrinología Molecular, Universidad Francisco de Vitoria, E-28223 Madrid, Spain; 5Centro de Investigación Biomédica en Red de Cáncer (CIBERONC), Instituto de Salud Carlos III, 28029 Madrid, Spain

**Keywords:** RAS, APT-1, VEGF-B, thyroid cancer

## Abstract

**Simple Summary:**

*RAS* mutations occur frequently in thyroid tumors, but the extent to which they are associated to tumor aggressiveness is still uncertain. HRAS proteins occupy different subcellular localizations, from which they regulate distinct biochemical processes. Herein, we demonstrate that the capacity of HRAS-transformed thyroid cells to extravasate and invade distant organs is orchestrated by HRAS subcellular localization, by a mechanism dependent on VEGF-B secretion. Interestingly, aggressiveness inversely correlates with tumor size. Moreover, we have identified the acyl protein thioesterase APT-1, a regulator of HRAS sublocalization, as a determinant of thyroid tumor growth versus dissemination. As such, alterations in APT-1 expression levels can dramatically affect the behavior of thyroid tumors. In this respect, APT-1 levels could serve as a biomarker that may help in the stratification of *HRAS* mutant thyroid tumors based on their aggressiveness.

**Abstract:**

*RAS* mutations are the second most common genetic alteration in thyroid tumors. However, the extent to which they are associated with the most aggressive phenotypes is still controversial. Regarding their malignancy, the majority of *RAS* mutant tumors are classified as undetermined, which complicates their clinical management and can lead to undesired under- or overtreatment. Using the chick embryo spontaneous metastasis model, we herein demonstrate that the aggressiveness of HRAS-transformed thyroid cells, as determined by the ability to extravasate and metastasize at distant organs, is orchestrated by HRAS subcellular localization. Remarkably, aggressiveness inversely correlates with tumor size. In this respect, we also show that RAS site-specific capacity to regulate tumor growth and dissemination is dependent on VEGF-B secretion. Furthermore, we have identified the acyl protein thioesterase APT-1 as a determinant of thyroid tumor growth versus dissemination. We show that alterations in APT-1 expression levels can dramatically affect the behavior of thyroid tumors, based on its role as a regulator of HRAS sublocalization at distinct plasma membrane microdomains. In agreement, APT-1 emerges in thyroid cancer clinical samples as a prognostic factor. As such, APT-1 levels could serve as a biomarker that could help in the stratification of *HRAS* mutant thyroid tumors based on their aggressiveness.

## 1. Introduction

RAS family GTPases, HRAS, NRAS, and KRAS cycle between an inactive, GDP-bound and an active, GTP-bound state, thereby acting as molecular switches in the relay of signals that orchestrate key cellular functions such as proliferation, differentiation, and survival. *RAS* is one of the gene families most frequently mutated in human cancer. About one-third of human tumors harbor mutations that lock RAS proteins in a constitutively active state, acting as drivers at the initial stages of carcinogenesis and as promoters of tumor dissemination [1].

To be functional, RAS proteins must be attached to the cytoplasmic leaflet of the plasma membrane (PM). This is achieved by posttranslational modifications at the C-terminus, that include farnesylation, proteolysis, carboxymethylation and, in the case of H and NRAS, palmitoylation (for an extensive review, see [2]). Largely as a consequence of their distinct posttranslational modifications, RAS isoforms occupy different microlocalizations at the PM, characterized by distinct biochemical composition and physical–chemical properties: KRAS is preferentially localized in the detergent-soluble fraction, also known as disordered membrane (DM), while H and NRAS are mainly present at detergent-insoluble lipid rafts (LR) [3,4,5,6,7] (for an extensive review, see [8]). In these microenvironments, RAS proteins are subject to site-specific regulatory events [8], differentially engage effector molecules [9,10], and switch on distinct transcriptional programs [11,12] leading to diverse biological outcomes [9,10,13,14,15]. Thus, space can introduce variability in RAS signals, depending on the availability, abundance, and functionality of regulators and effectors at different locations.

Furthermore, RAS localization is not static. RAS traffics between LR and DM depending on its activation [3] and palmitoylation status [16]. Palmitoylation is also the main orchestrator of HRAS and NRAS transit between the PM and endomembranes, like the Golgi complex (GC) and the endoplasmic reticulum (ER). Once at the PM, HRAS and NRAS are depalmitoylated therein and trafficked back to the GC, where a new palmitoylation process will deliver them, again, to the PM [17,18]. Depalmitoylation is undertaken by acyl-thioesterases such as APT-1, a cytosolic enzyme that is active on HRAS [19,20]. As such, APT-1 also regulates HRAS traffic between LR and DM [16].

Thyroid cancer is the most common endocrine malignancy worldwide, accounting for ~2.1% of all cancer diagnoses, and its incidence continues to rise, although mortality, about 5%, has not changed significantly over the past fifty years [21,22]. Thyroid tumors usually arise as nodules, which are evaluated and classified into six categories, following the 2017 Bethesda System for Reporting Thyroid Cytopathology, that predict their risk of malignancy and subsequent management [23]. Around 95% of cases are classified as differentiated thyroid cancer (DTC). Papillary thyroid cancer (PTC) and follicular thyroid cancer (FTC) are included in this category. Following the 2015 American Thyroid Association (ATA) guidelines, patients with a >1 cm nodule diagnosed as DTC are recommended surgical removal as primary treatment [24]. However, since thyroid surgery is not without hazards, major challenges for clinicians are not to overtreat patients at low risk and to identify those patients with a high-risk disease in order to treat them fast and more aggressively [25].

*RAS* mutations are the second most common genetic alteration in thyroid tumors [26]. However, the extent to which they are associated to the most malignant phenotypes is still controversial. Several studies have related mutant *RAS* with distant metastasis [27,28] and high mortality rates [29,30]. The association to malignancy also varies depending on the mutant isoform, being greatest with *HRAS* and the least with *KRAS* [31]. However, *RAS* mutations are also common in follicular adenomas that seldom progress to cancer [32]. In this line, in a series of *RAS*-positive thyroid nodules, 3% were malignant, 3% benign, and 94% undetermined, with the latter falling into the Bethesda III–IV categories and may or may not progress to invasive tumors [33]. As such, in the majority of the cases, *RAS* mutant tumors are difficult to manage, which can lead to an undertreatment of patients with highly aggressive tumors, or to an overtreatment of those with more benign forms [24,34]. Nowadays, no test is available to discriminate between high- or low-risk *RAS* mutant tumors and predict how they are going to progress.

Herein, we show that HRAS-transformed thyroid cells yield primary tumors of different size depending on the subcellular localization from which HRAS signals emanate. Remarkably, tumor size is inversely correlated with the ability to extravasate and colonize distant organs. We also present data showing that HRAS capacity to regulate tumor growth/dissemination is dependent on vascular endothelial growth factor B (VEGF-B) secretion. Importantly, we demonstrate that up- or downregulation of APT-1 levels impacts on HRAS distribution at different plasma membrane microdomains and, consequently, on the capacity of HRAS-transformed thyroid cells to proliferate in situ versus disseminate. As such, our results unveil APT-1 as a potential marker of the aggressiveness of HRAS mutant thyroid tumors.

## 2. Results

### 2.1. RAS Sublocalization Differentially Affects Transformed Thyroid Cells Proliferation and Dissemination

To gain a first insight into how RAS sublocalization impacts on thyroid cells’ neoplastic behavior, we utilized rat PCCL3 thyroid follicular cells, that, though spontaneously immortalized, do not display tumorigenic capacity per se [35]. From these cells, we generated cell lines stably expressing HRAS^V12^ (total) and the same oncoprotein targeted to defined subcellular localizations by the aid of N-terminally fused tethers: LCK-HRAS^V12^ targeted to lipid rafts (LR); CD8-HRAS^V12^ to disordered membrane (DM); M1-HRAS^V12^ to endoplamic reticulum (ER); and KDELr-HRAS^V12^ to the Golgi complex (GC) [9,14] (Figure 1A; Appendix A). In these cell lines, we evaluated the ability of HRAS^V12^ to regulate in vitro proliferation depending on its localization, finding no significant differences compared to parental cells (Figure 1B). Likewise, we also monitored the HRAS^V12^ site-dependent capacity for promoting in vitro cellular migration using Transwell assays. In this case, we found that cells expressing HRAS^V12^ at DM and GC displayed much higher migration rates than those where HRAS^V12^ was localized at LR and ER (Figure 1C).

In the same vein, we extended our analyses to in vivo settings and utilized chicken embryo chorioallantoic membrane (CAM) xenografts [36,37] as an animal model that faithfully recapitulates most characteristics of carcinogenesis, such as tumor growth, intravasation, and distant colonization. The different cell lines were grafted on the embryo CAM and allowed to grow for seven days, after which intravasation, evaluated by the presence of invading cells in the distal CAM (Appendix A), and the occurrence of lung metastases, were analyzed and quantified by qPCR. It was found that cells harboring HRAS^V12^ at ER and LR generated bigger tumors than those in which HRAS^V12^ signaled from DM and GC (Figure 2A and Appendix A). Interestingly, the effect was reversed when intravasation and lung metastasis were evaluated, as cells expressing HRAS^V12^ at DM and GC displayed greater intravasation and metastatic capacity than those with HRAS^V12^ at ER and LR (Figure 2B,C and Appendix A). Similar results were obtained for cells with site-specific expression of NRAS^V12^ (Appendix A), demonstrating that such effect was independent of the isoform.

The possibility existed that our observations are an artifact resulting from the overexpression of artificially tethered HRAS constructs. Thus, we investigated if the site-specific activation of endogenous RAS yielded similar results. To this end, we used an engineered exchange factor made up of RASGRF1 CDC25 domain fused to the aforementioned tethers, whereby it constitutively activated RAS only at specific sublocalizations [12,38]. Remarkably, these constructs yielded tumors of comparable size to those generated by HRAS^V12^ with identical spatial differences: those in which RAS was activated at ER and LR were bigger than those yielded by RAS from DM and GC (Figure 2D). However, the extravasation and metastatic potential of these tumors was remarkably lower than those driven by HRAS^V12^, though the behavior as defined by the different sublocalizations persisted: DM and GC were more metastatic than ER and LR (Figure 2E,F).

Next, we evaluated if these observations persisted in physiological settings. For this purpose, we utilized the HRAS mutant cell lines C643 and HTH83, derived from thyroid carcinomas [39,40]. In these cells, we determined HRAS distribution at the plasma membrane by fractionation analyses, and found that HRAS localized mostly at LR in C643 cells, whereas it segregated at DM in the case of HTH83 (Figure 3A and Appendix A), probably as a consequence of C643 cells expressing more APT-1 than HTH83 (Figure 3B), since high APT-1 activity promotes HRAS accumulation at LR [16]. When the carcinogenic potential of these cell lines was tested in the chick CAM model, it was found that C643 cells generated bigger tumors than HTH83 (Figure 3C). Contrarily, HTH83 cells displayed greater intravasation and lung colonization capacity (Figure 3D,E). These data fully recapitulate the results obtained using the tethered HRAS constructs and demonstrate that HRAS carcinogenic behavior is dependent on its subcellular localization that, in thyroid cells, antagonistically regulates tumor growth and dissemination.

### 2.2. HRAS Regulates Tumor Behavior via VEGF-B Secretion

It was of interest to unravel the mechanism whereby HRAS regulates the antagonistic behavior of thyroid tumor cells with respect to their growth and dissemination capabilities. In the course of the histological analyses of CAM tumors, it was observed that tumor cells exhibited empty vacuoles reminiscent of lipid droplets dissolved by the histological processing. To ascertain this point, we performed Oil Red staining which, indeed, revealed their presence in tumor cells. These were particularly prominent in those tumors driven by ER and LR HRAS^V12^ (Figure 4A and Appendix A).

VEGF-B is a poorly angiogenic member of the VEGF family of growth factors [41]. It has been associated to energy metabolism as an inducer of long-chain fatty acid uptake and storage [42]. Thus, we hypothesized that this molecule could be responsible for the lipid accumulation in tumor cells. To test this, we evaluated VEGF-B levels in the different site-specific HRAS^V12^ cell lines, where we found that those expressing ER and LR HRAS^V12^ displayed the highest levels of VEGF-B mRNA and protein (Figure 4B), in full agreement with their greatest droplet content. Likewise, VEGF-B levels were much higher in C643 cells harboring HRAS in LR than in HTH83 with HRAS at DM (Figure 4C). Noticeably, we had observed that ER and LR HRAS^V12^ also yielded the biggest tumors, just as C643 cells (Figure 2A and Figure 3C). In light of this, we asked whether VEGF-B enhanced tumor growth. For this, we grafted cells harboring DM and GC HRAS^V12^ that originally yield small tumors (Figure 2A) and allowed them to grow in the presence of exogenously added VEGF-B. It was found that VEGF-B substantially stimulated tumor growth in both cell lines (Figure 5A and Appendix A) but, at the same time, it reduced their propensity to intravasate (Figure 5B) and to metastasize (Figure 5C).

In the same vein, we tested if reducing VEGF-B levels had an inhibitory effect on tumor growth. For this purpose, VEGF-B expression was downregulated by shRNA interference in cells harboring ER and LR HRAS^V12^ (Figure 5D). When the behavior of these cells was analyzed in the chick CAM model it was found that, indeed, diminished VEGF-B expression impaired tumor growth (Figure 5E) while it enhanced the ability of these cells for intravasating (Figure 5F) and colonizing distant organs (Figure 5G). Overall, these results demonstrate that HRAS controls the growth and dissemination of thyroid tumors through an autocrine regulatory loop mediated by VEGF-B.

### 2.3. APT-1 Overexpression Correlates with Better Prognosis in Thyroid Tumors

We have previously demonstrated that APT-1 regulates HRAS distribution between LR and DM microdomains in the plasma membrane [16]. Since thyroid tumors behave differently depending on whether HRAS signals from DM or LR, it was of interest to understand how APT-1 levels related to tumor evolution. In the cBioPortal for Cancer Genomics database (TCGA), we selected a cohort of 507 samples of thyroid tumors, of which 77.3% correspond to papillary carcinomas, 20.7% follicular carcinomas, and 2% to poorly differentiated tumors [43], where we looked for genetic alterations in APT-1 (lysophospholipase 1, LYPLA1) and their association to those in HRAS and NRAS. It was found that 7% of the cases exhibited alterations in APT-1, the vast majority (28/34) resulting in overexpression. Noticeably, 15/34 coincided with RAS gain-of-function alterations (Appendix A).

When we looked at the evolution of tumors harboring RAS mutations with respect to the APT-1 status, it was found that while patients harboring tumors without APT-1 alterations had a 14-year survival rate of 80%, all of the cases showing APT-1 overexpression had survived after the same period (Figure 6). Unfortunately, the number of patients showing APT-1 downregulation (*n* = 4) was too low for a significant analysis. These clinical data are in full agreement with our previous results showing that high APT-1 levels result in HRAS confinement to LR, resulting in bigger tumors though with reduced metastatic potential.

### 2.4. APT-1 Levels Determine Thyroid Tumor Behavior

Accordingly, we tested whether alterations in APT-1 levels would impact on the carcinogenic properties of thyroid tumor cells. As shown above, C643 express high levels of APT-1 (Figure 3B), so we generated clones in which these were stably reduced by shRNA interference (APT-1 KD) (Figure 7A and Appendix A). As a consequence, HRAS that was found in LR in parental cells (Figure 3A) changed its localization to DM (Figure 7B). When the behavior of these cells was analyzed in the chick CAM model, it was found that the tumors generated by the APT-1 KD cells were smaller than those derived from C643 parental cells (Figure 7C). However, the APT-1 KD cells exhibited an enhanced intravasation and metastatic capacity (Figure 7D,E).

In a similar fashion, in HTH83 cells originally expressing low levels of APT-1 (Figure 3B), we ectopically expressed APT-1 to create overexpressing clones (APT-1 OE) (Figure 7F). In these, HRAS partially shifted to LR (Figure 6G), unlike parental cells, where it was exclusively found in DM (Figure 3A). When tested in the chick CAM model, HTH83 APT-1 OE cells produced bigger tumors than their parental counterparts (Figure 7H). Contrarily, these cells displayed a reduced ability for intravasation and lung colonization (Figure 7I,J).

Overall, these data demonstrate that the status of APT-1 expression, as a consequence of its influence on HRAS sublocalization at different plasma membrane microdomains, markedly impacts on thyroid tumor size and on thyroid tumor cells’ ability for intravasation and distant colonization.

## 3. Discussion

Although, in the last decade, considerable efforts have been devoted to correlate *RAS* mutations to the clinical management of thyroid cancer, most *RAS*-positive tumors are still classified as indeterminate using the Bethesda System, with some cases being highly aggressive and others showing an indolent behavior [44]. Drugs like the farnesyl transferase inhibitor tipifarnib effectively target HRAS, evoke beneficial cellular responses, and extend survival in thyroid cancer animal models, pointing to this GTPase as a promising therapeutic target [45]. In this study, we introduce subcellular localization, an erstwhile unconsidered factor, as a determinant of the dissemination potential of thyroid tumors driven by either HRAS or NRAS oncoproteins. We demonstrate that H/NRAS signals emanating from LR or ER yield big tumors with a reduced propensity to disseminate in comparison to tumors driven by H/NRAS signals coming from DM or GC, which are smaller but with a greater capacity to intravasate and metastasize in distant organs.

Interestingly, we have found that this tumor growth/dissemination antagonism, as regulated by RAS sublocalization, is orchestrated by a VEGF-B autocrine loop in such a way that high VEGF-B levels promote big tumors with reduced propensity for dissemination, whereas curtailing VEGF-B secretion results in smaller tumors, though with enhanced metastatic potential. Although VEGF-B overexpression has been detected in some types of tumors [46], its role in tumorigenesis is still obscure. Even though VEGF-B is not an angiogenic factor, it antagonizes the formation of abnormal, permeable vessels as induced by VEGF-A [47]. As such, it can be envisioned that by counteracting the establishment of leaky vasculature that facilitates cellular intravasation, VEGF-B would diminish metastatic dissemination and favor tumor growth by preventing cells from escaping and retaining them at the bulk of the tumor. In this respect, it is worth indicating that lenvantinib and other tyrosine kinase inhibitors utilized for the treatment of advanced thyroid cancer are known to inhibit VEGF receptors 1–3 [48], highlighting the importance of this signaling axis in thyroid neoplasia.

We had previously demonstrated that the sublocalization from which oncogenic RAS signals originate determines its transforming potential and orchestrates tumor behavior [9,13,14]. With respect to RAS signals emanating from the GC, it is particularly striking that in the case of melanoma, these exert antineoplastic effects [14]. Contrarily, as we show here, they produce highly invasive tumors in the thyroid. These antagonistic outcomes suggest that mutant RAS oncogenicity is both site-specified and context-dependent, varying dramatically depending on the cell type or tissue where it occurs. Thus, in the absence of a location-specific RAS signature, the availability of a biomarker whereby RAS localization can be inferred would be most helpful as a predictor of RAS oncogenic potential.

In this respect, herein, we unveil APT-1 as a potential indicator for favorable prognosis in thyroid papillary carcinomas. Clinical data obtained from TCGA, mostly corresponding to papillary thyroid carcinomas, shows that patients with high APT-1 levels are associated with better overall survival rates. In light of our data, this can be explained by APT-I overexpression promoting HRAS localization to LR where, as we demonstrate, HRAS generates tumors with a diminished propensity for metastasis. In this line, we also show that by altering APT-1 expression levels, thereby changing HRAS distribution between LR and DM microdomains, tumor growth and metastatic behavior can be dramatically modified. It is noteworthy that, according to the TCGA data, more than 50% of the cases with APT-1 overexpression do not coincide with RAS mutations. This could suggest that even altering wild-type RAS distribution could have an impact on thyroid tumor behavior. Studies on a broader panel of clinical cases will be necessary in order to reach a final conclusion in this respect.

Overall, contrarily to the prevailing concept that bigger tumors are more prone to metastasize, we herein demonstrate that HRAS/NRAS-driven thyroid tumors exhibit the opposite behavior. This is of concern since ATA guidelines recommend surgical removal only for DTC nodules bigger than 1 cm [24]. If HRAS/NRAS-positive, this would imply that, depending on RAS sublocalization, small though highly metastatic tumors could pass untreated. Due to the uncertain role of *RAS* mutations per se in the unfolding of thyroid carcinogenesis, new biomarkers such as APT-1 would be determinant for avoiding this undesired situation.

## 4. Materials and Methods

### 4.1. Cell Culture

PCCl3 cells, well differentiated rat follicular thyroid cells (RRID:CVCL_6712) were provided by Dr. Fusco (Istituto di Endocrinologia ed Oncologia Sperimentale-CNR, Dipartimento di Medicina Molecolare e Biotecnologie Mediche, Università degli Studi di Napoli “Federico II,” Naples, Italy) were grown in DMEM with 5% donor calf serum, supplemented with a six-hormone mixture (6H; 1 nm thyrotropin; 10 μg/mL insulin; 10 ng/mL somatostatin; 5 μg/mL transferrin; 10 nm hydrocortisone; and 10 ng/mL glycyl-l-histidyl-l-lysine acetate) all from Sigma, and penicillin–streptomycin (10,000 U/mL) (Life Technologies, Inc, Waltham, MA, USA). Human HRAS mutant thyroid anaplastic carcinoma cell lines HTH83 (RRID:CVCL_0046) and C643 (RRID:CVCL_5969) cells were kindly donated by Dr. N.E. Heldin (University of Uppsala, Uppsala, Sweden) and were grown in DMEM with 10% fetal bovine serum and penicillin–streptomycin (Life Technologies, Inc). *Mycoplasma* testing was undertaken every 6 months using commercial *Mycoplasma* testing kits (Biotools, Madrid, Spain). All cell lines used in this work have been authenticated every 12 months by short tandem repeat profiles using the Applied Biosystems Identifier kit in the Genomic Facility at IBBTEC. Where applicable, stable lines cells were generated by transfection with Lipofectamine (Invitrogen, NY, USA) and selected with 750 μg/mL G418. Transient transfections were performed with Lipofectamine^TM^ 3000 (Invitrogen).

### 4.2. Plasmids and siRNAs

Plasmids carrying the HRAS^V12^ mutants have been previously described [9]. The same epitopes and localization signals used for pCEFL-HA- HRAS^V12^ were used for the generation of M1-FLAG-CDC25, LCK-FLAG-CDC25, CD8-FLAG-CDC25, and KDEL-FLAG-CDC25 by cloning FLAG-CDC25 in the C-terminus of the different localization-targeting vectors as previously described [38]. Differences in MWs are due to the sizes of the tethering signals fused to either HRAS or CDC25. Specifically: M1, 7.2 kDa; LCK, 3.5 kDa; CD8, 45.2 kDa; KDELr, 44.4 KDa. All sequences were verified by DNA sequencing. Small interfering RNA (siRNA) against LYPLA1 was from Santa Cruz (sc-7763). APT-1 was amplified by reverse transcription-PCR and subcloned into pCEFL FLAG. A sense–antisense 19-base oligonucleotide targeting rat VEGF-B separated by a hairpin was cloned into pSUPER retro. The sh RNA sequence used was 5′-GATCCCCAGCCAACGTGGTAACCAGCTTTTCAAGAGAAAGCTGGTTACCACGTGGCTTTTT-3′.

### 4.3. Immunoblotting

Cell lysis was performed as described [49]. Protein concentration was determined by Bradford (Bio-Rad Laboratories, Hercules, CA, USA). Samples were separated by SDS-PAGE and transferred to nitrocellulose membranes (Bio-Rad Laboratories). The following antibodies were used: HRAS (Abcam Cat# ab97488, RRID:AB_10680439); VEGF-B (Abcam Cat# ab51867, RRID:AB_2304198); transferrin receptor (Abcam Cat# ab84036, RRID:AB_10673794); Caveolin (BD Biosciences Cat# 610684, RRID:AB_398009); APT-1 (LYPLA1) (Novus Cat# H00010434-M05, RRID:AB_1146125); tubulin (Sigma-Aldrich Cat# T8328, RRID:AB_1844090).

### 4.4. Proliferation and Migration Analyses

Proliferation assays were performed using AlamarBlue Cell Viability Reagent (Thermo Fisher, Waltham, MA, USA). Around 6000 cells/well were plated in a 96-well plate in 100 μL medium and starved for 12 h. After that time, 10 μL of AlamarBlue Reagent was added and incubated in the dark at 37 ℃ for 12 h. Absorbance was read at 540 and 620 nm. Cell migration was examined in Transwell cell culture chamber filters (8 μm pore) (Corning). Cells were seeded at 5 × 10^4^ cells in DMEM with 0.2% FBS. Following 48 h incubation, the invading cells were fixed and analyzed by fluorescence microscopy and counted. Images were processed and analyzed using Fiji Image.

### 4.5. Chick Embryo Spontaneous Metastasis Model

Basically conducted as previously described [36]. Briefly, chick embryos (Granja Gibert, Tarragona, Spain) were allowed to develop at 37 °C in a humidified incubator. After 10 days, 10^6^ thyroid cells were grafted through a window opened in the eggshell onto the embryo’s CAM. On day 7, primary tumors were removed and weighed, and portions of the distal CAM and lungs were excised and analyzed by qPCR to determine actual numbers of human (Alu sequences) or rat cells in the chicken tissues. The CAM chick metastasis embryo assay does not require administrative procedures for obtaining ethics committee approval for animal experimentation because the chick embryo is not considered as a living animal until day 17 of development. The CAM is not innervated, and experiments were terminated before the development of centers in the brain associated with pain perception, making this a system not requiring animal experimentation permissions. All experiments were performed according to the national guidelines for animal care in accordance with the European Union Directive.

### 4.6. DNA/RNA Extraction from Chick Embryos and Quantification

To extract genomic DNA from the different organs we used the Gentra PureGen Tissue Kit from QIAGEN (Germantown, MD, USA), following manufactures instructions. Briefly, organs were harvested in lysis buffer, homogenized, and incubated at 65 ℃ o/n. The next day, protein precipitation buffer was added. Samples were centrifuged and the supernatant was treated with isopropanol for DNA precipitation. For tumor RNA extraction and quantification, frozen tumors were minced on dry ice and RNA was extracted using the RNeasy Mini Kit (QIAGEN) following manufacturer’s instructions. cDNA was synthesized using iScript™ Reverse Transcription Supermix (BioRad). After resuspension, nucleic acid contents were quantified used NanoDropTM 2000c (Thermo Fisher, Waltham, MA, USA).

### 4.7. Real Time qPCR

In order to detect metastatic cells in the chicken tissues, primers for rat β-actin or for human Alu sequences (Sigma, Saint Louis, MO, USA) were utilized. DNA (30 ng) was used for PCR following the manufacturer’s instructions (PowerUp SYBR Green Master Mix, Thermo Fisher). PCR conditions were the following: 4 min/ 95 ℃ followed by 40 cycles of 30 s at 95 ℃ to denature DNA, 63 ℃/30 s for primer annealing, and 30 s at 72 ℃ for amplification. The primers used were:

Forward rat primer (light myosin chain) 5′-CAAAAATGGAGCTGCGCAGGC-3′,

Reverse rat primer (light myosin chain) 5′-CGCCAGCTGGTGGGGATTTTA-3′;

Forward Alu human primer 5′-ACGCCTGTAATCCCAGGACTT-3′,

Reverse Alu human primes 5′-TCGCCCAGGTGGCTGGGGCA-3′,

Forward VEGF-B rat primer 5′-GATCCAGTACCCGAGCAGTCA-3′,

Reverse VEGF-B rat primer 5′-TGGCTTCACAGCACTCTCCTT-3′.

The number of human or rat cells were determined by the triplicate Ct values against a standard curve generated from a specific known number of human or rat cells (100, 1000, and 10,000 cells).

### 4.8. Plasma Membrane Fractionation in Sucrose Gradients

Cells were collected and treated as previously described [9]. Briefly, cells were resuspended in 25 mM Tris, pH 7.4, 150 mM NaCl, 5 mM EDTA, 0.25% Triton X-100 plus protease inhibitors cocktail (1 μg/mL). Lysates were set at a sucrose concentration of 45%. Layers of 3.4 mL of 35% sucrose and 1 mL of 16% sucrose were sequentially overlaid and centrifuged for 18 h at 41,000 rpm (MLS-50 rotor, Beckman). Twelve 0.4 mL fractions were collected and resuspended directly into SDS–PAGE sample buffer for analysis by immunoblotting.

### 4.9. Statistical Analyses

All statistical data were analyzed and compared for statistically significant differences by two-tailed unpaired Student’s *t*-test or Mann–Whitney tests (GraphPad Prism 8 (GraphPad Software, San Diego, CA, USA).

## 5. Conclusions

In summary, we demonstrate that HRAS-driven thyroid tumors potential for dissemination is highly dependent on the sublocalization from which HRAS signals emanate according to a mechanism mediated by VEGF-B secretion. Strikingly, while the prevailing concept is that a tumor dissemination capacity increases with its size, we herein demonstrated that HRAS-driven thyroid tumors exhibit the opposite behavior.

## Figures and Tables

**Figure 1 cancers-12-02588-f001:**
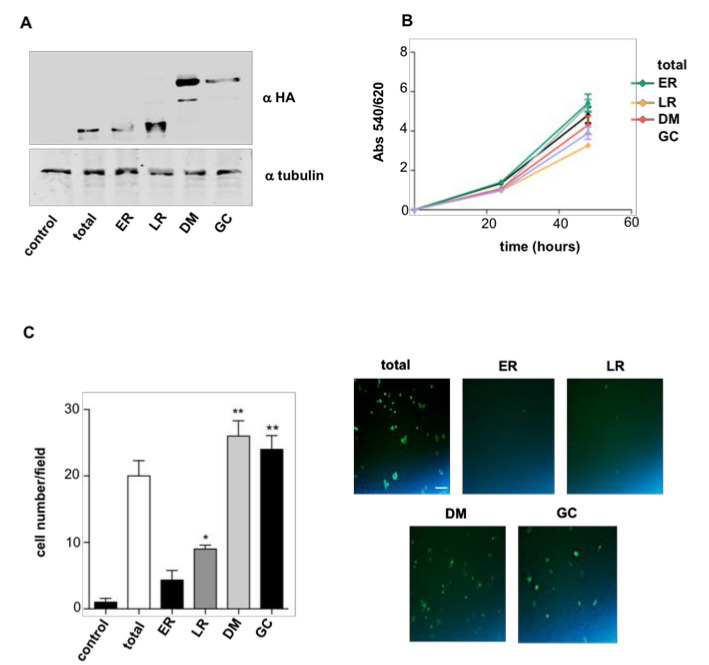
HRAS site-specific effects on cellular proliferation and migration. (**A**) Expression levels of the targeted RAS proteins in stably transfected PCCL3 as determined by anti-HA immunoblotting. Cells were transfected with empty vector (control); HRAS^V12^ (total); M1-HRAS^V12^ (endoplasmic reticulum (ER)); LCK-HRAS^V12^ (LR); CD8-HRAS^V12^ (disordered membrane (DM)); and KDELr-HRAS^V12^ (Golgi complex (GC)). (**B**) Cellular proliferation rate of the aforementioned cell lines. (**C**) Transwell migration assay for the aforementioned cell lines. Left panel: data show mean ± SD of cells per field of three independent experiments. * *p* < 0.05, ** *p* < 0.01 by two- tailed unpaired Student *t*-test. Right panel: Caption of migrated cells through 8 μm pore transwell after 24 h. Scale bar = 50 μm.

**Figure 2 cancers-12-02588-f002:**
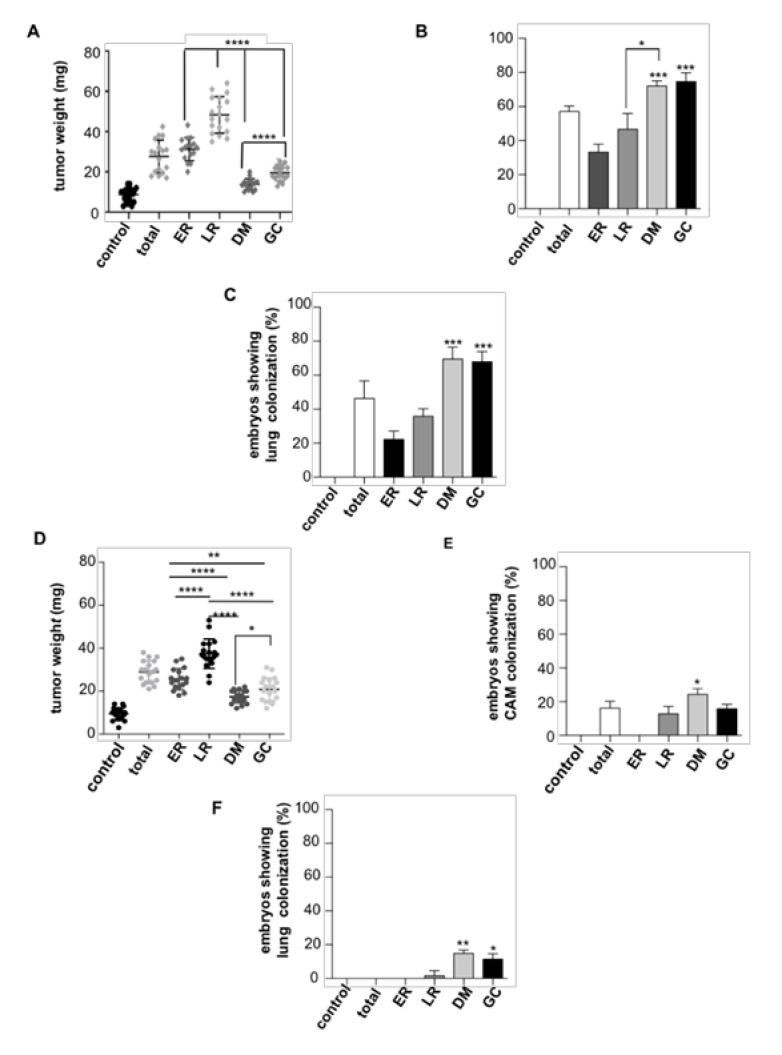
HRAS site-specific effects on tumor growth and dissemination. Cells (10^6^) from each PCCL3-derived cell line stably expressing the indicated site-specific HRAS^V12^ constructs were grafted on chick embryos and allowed to grow for seven days. (**A**) Tumor size. (**B**) Intravasation, expressed as the % of chicken displaying distal chorioallantoic membrane (CAM) colonization. (**C**) Distant metastases, expressed as the % of chicken displaying lung colonization. (**D**–**F**) As before, but refer to PCCL3-derived cell lines expressing the indicated site-specific CDC25 constructs. Data show mean ± SD (**A**,**D**); mean ± SEM (**B**,**C**,**E**,**F**) from three independent experiments using 9-15 embryos per case. * *p* < 0.05, ** *p* < 0.01 *** *p* < 0.001 and **** *p* < 0.0001 by two-tailed unpaired Student *t*-test.

**Figure 3 cancers-12-02588-f003:**
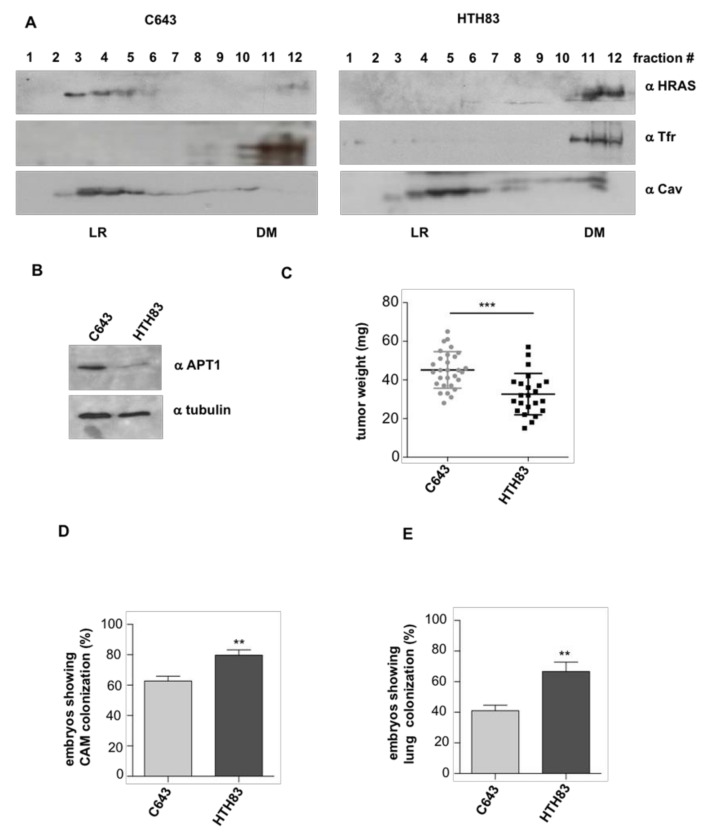
HRAS distribution in plasma membrane (PM) microdomains is related to tumor behavior. (**A**) Endogenous HRAS localization at LR and DM fractions in the indicated cell lines. Caveolin-1 (Cav) and transferrin receptor (Tfr) serve as specific markers for their respective microdomains. (**B**) APT-1 expression levels in the indicated cell lines. (**C**) Size of the tumors generated by the indicated cell lines when grafted (10^6^ cells) on chick embryos. (**D**) Intravasation, expressed as the % of chicken displaying distal CAM colonization. (**E**) Distant metastases, expressed as the % of chicken displaying lung colonization. Data show mean ± SD (**C**); mean ± SEM (**D**,**E**) from three independent experiments using 8–16 embryos per case. ** *p* < 0.01 and *** *p* < 0.001 by two-tailed unpaired Student *t*-test.

**Figure 4 cancers-12-02588-f004:**
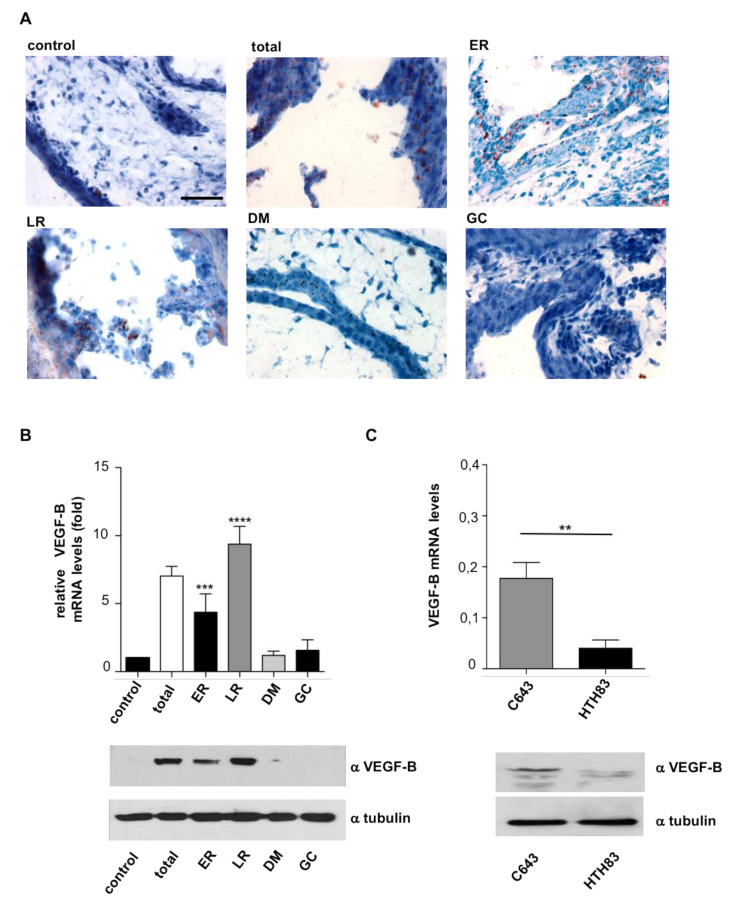
VEGF-B levels in relationship with HRAS sublocalization. (**A**) Lipid accumulation as unveiled by Oil Red staining in sections from tumors generated in chick embryos by the indicated HRAS^V12^ site-specific constructs. Scale bar = 100 μm. (**B**) VEGF-B mRNA and protein levels in the tumors generated by the indicated HRAS^V12^ site-specific constructs relative to those in control cells. (**C**) VEGF-B mRNA and protein levels in tumors generated by the indicated cell lines. Data show mean ± SEM from three independent experiments using 8–10 embryos per case. ** *p* < 0.01 and *** *p* < 0.001, **** *p* < 0.0001 by two-tailed unpaired Student *t*-test.

**Figure 5 cancers-12-02588-f005:**
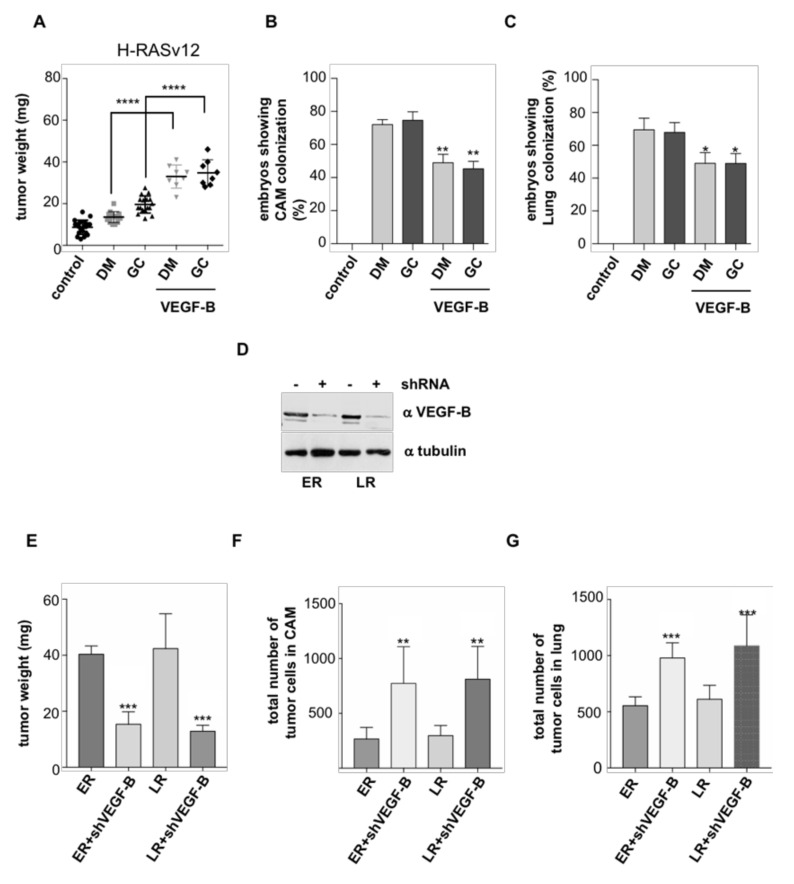
Changes in VEGF-B levels alter tumor behavior. Cells (10^6^) of the indicated HRAS^V12^ site-specific cell lines were grafted on chick embryos. Where indicated, VEGF-B (10 nM) was added every two days beginning from day 2. After 8 days, primary tumors were collected and tissues analyzed by qPCR. (**A**) Size of the tumors. (**B**) Intravasation, expressed as the % of chicken displaying distal CAM colonization. (**C**) Distant metastases, expressed as the % of chicken displaying lung colonization. (**D**) shRNA-mediated knockdown of VEGF-B levels in PCCL3 cell lines expressing HRAS^V12^ at LR and ER, as indicated. (**E**) Size of the tumors, (**F**) intravasation, and (**G**) distant metastases, generated by the indicated cell lines in which VEGF-B expression has been downregulated. Data show mean ± SD (**A**); mean ± SEM (**B**–**E**) from three independent experiments using 5–10 embryos per case. * *p* < 0.05, ** *p* < 0.01, *** *p* < 0.001 and, **** *p* < 0.0001 by two-tailed unpaired Student *t*-test.

**Figure 6 cancers-12-02588-f006:**
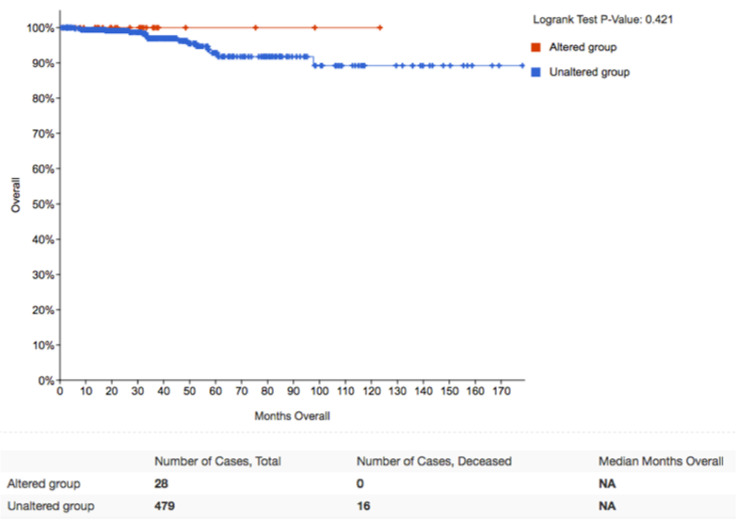
APT-1 levels determine the clinical outcome of thyroid tumors. Kaplan–Meier survival curve corresponding to genetic alterations in APT-1 (LYPLA1), HRAS, and NRAS in papillary thyroid carcinomas. Obtained from a cohort of 507 patients at cBioPortal for Cancer Genomics database (TCGA).

**Figure 7 cancers-12-02588-f007:**
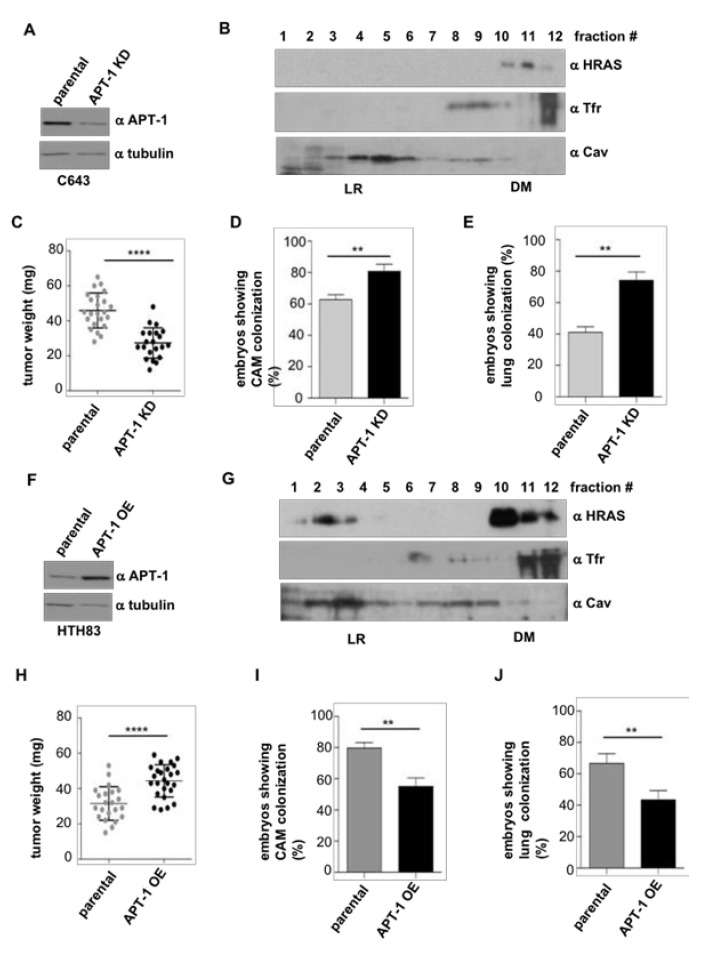
Alterations on APT-1 levels determine thyroid tumor behavior. (**A**) APT-1 levels in C643 cells after APT-1 shRNA-mediated knockdown (APT-1KD) compared to parental cells. (**B**) Endogenous HRAS localization at LR and DM fractions of C643 APT-1 KD cells. (**C**) Size of the tumors, (**D**) intravasation, and (**E**) distant metastases in APT-1 KD cells compared to the parental C643 line. (**F**) APT-1 levels in HTH83 cells after ectopic overexpression (APT-1OE), compared to parental cells. (**G**) Endogenous HRAS localization at LR and DM fractions of HTH83 APT-1 OE cells. (**H**) Size of the tumors; (**I**) Intravasation and (**J**) Distant metastases, in APT-1 OE cells compared to the parental HTH83 line. Data show mean ± SD (**C**,**H**); mean ± SEM (**D**,**E**,**I**,**J**) from three independent experiments using 6–12 embryos per case. ** *p* < 0.01 and **** *p* < 0.0001 by two-tailed unpaired Student *t*-test.

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
