# Peer review of "RAS Subcellular Localization Inversely Regulates Thyroid Tumor Growth and Dissemination"

_cancers, 2020, doi:10.3390/cancers12092588_

Round 1

Reviewer 1 Report

In this paper the authors present the association of RAS cellular localization and clinical outcomes of thyroid cancer.

This is an interesting study and as the authors state, RAS is an interesting driver in thyroid cancer being associated with many benign follicular nodules and also aggressive metastatic disease when it is associated with other associated mutations such as TERT promoter and TP53 mutations. In addition most thyroid nodules with an atypical biopsy (B3 or B4) will often have a RAS mutation making a better understanding of these mutations an opportunity for translation to improve patient care and decision making.

The authors show that in rat PCCL3 thyroid cells with a HRASG12V mutation there is no difference in cellular proliferation dependent on RAS localisation but RAS localised to the disordered membrane – DM or golgi complex - GG had higher migration than those at lipid rafts LR and endoplasmic reticulum ER.

The authors used these cells in embryo chorioallantoic membrane (CAM) xenografts and found that HRAS localised at the ER and LR had bigger tumours, while HRAS at DM and GG had greater metastasis. The authors then used the human HRAS mutant cell lines – C643 and HTH83 – 2 lines derived from anaplastic thyroid cancers with G13R HRAS and Q61R HRAS mutation. The authors find that HRAS was more localised in LR in C643 cells and DM in HTH83 cells and explore the relationship of localisation to APT-1, VEGF-B and present some human data from the TCGA.

The abstract needs to specify the authors results are on HRAS and displayed in cell lines and a chicken embryo model of thyroid cancer.

In the introduction there is an error in the mention of the Bethesda System for Reporting Thyroid Cytopathology which has six categories and not seven.

I feel the authors conclusion for their study with the HRAS cell lines is too strong and that they can only associate that carcinogenic behavior of HRAS is dependent on subcellular localistaion and not that ‘thyroid cells carcinogenic behavior is orchestrated depending on HRAS localization’ which would need knockout and rescue experiments to support these stronger conclusion.

The data for APT-1 accumulation is interesting but would be helped if the authors explain why those chose to investigate the level of expression of this enzyme, did the authors test for any of members of the RAS pathways which could also be influencing this result? This information would be useful to mention in the manuscript and include in the supplementary material.

The results for VEGFB is also of interest, especially as many TKIs used in advanced thyroid cancer such as Lenvatinib are thought to mainly act through inhibition of VEGF which could be explored further in the discussion, do the authors have any protein expression results to support the differences in mRNA expression?

In figure 6A the text labelling each group has been cut off and the figure is not of high enough resolution to see clearly, are the authors showing here that APTs had higher RNA expression from the TCGA study?

The authors should report the other common drivers such as BRAF and known prognostic mutations such as TERT to this figure.

I cannot see how the authors can reach the conclusion that higher APT expression leads to improved survival, the most which could be said is that is it associated with a better overall survival but no P-value is reported, as nearly all the patients in the TCGA series had well differentiated PTCs it is likely many of the deaths reported are not due to thyroid cancer and for thyroid cancer deaths they are likely related to known drivers of aggressive disease, such as BRAF and TERT.

Inhibitors of HRAS of tipifarnib are available as demonstrated in this paper https://cancerres.aacrjournals.org/content/canres/78/16/4642.full.pdf and should be discussed by the authors for further study of their model.

In the conclusion the authors extend there results to NRAS but only one set of experiments use NRAS so I do not see how this can be supported.

Author Response

We thank the reviewer for his constructive suggestions

The abstract needs to specify the authors results are on HRAS and displayed in cell lines and a chicken embryo model of thyroid cancer.

Agree, we have rewritten the abstract accordingly: “Using the chick embryo spontaneous metastasis model, herein we demonstrate that the aggressiveness of HRAS-transformed thyroid cells, as determined by the ability to extravasate and metastatize at distant organs, is orchestrated by HRAS subcellular localization”.

In the introduction there is an error in the mention of the Bethesda System for Reporting Thyroid Cytopathology which has six categories and not seven.

Corrected

I feel the authors conclusion for their study with the HRAS cell lines is too strong and that they can only associate that carcinogenic behavior of HRAS is dependent on subcellular localistaion and not that ‘thyroid cells carcinogenic behavior is orchestrated depending on HRAS localization’ which would need knockout and rescue experiments to support this stronger conclusion.

Agree, we have rewritten it: “….and demonstrates that HRAS carcinogenic behavior is dependent on its subcellular localization that, in thyroid cells, antagonistically regulates tumor growth and dissemination”.

The data for APT-1 accumulation is interesting but would be helped if the authors explain why those chose to investigate the level of expression of this enzyme, did the authors test for any of members of the RAS pathways which could also be influencing this result? This information would be useful to mention in the manuscript and include in the supplementary material.

In the Introduction we mention: “Depalmitoylation is undertaken by acyl-thioesterases such as APT-1, a cytosolic enzyme that is active on HRAS [19, 20]. As such, APT-1 also regulates HRAS traffic between LR and DM [16]”. And in the Results section prior to describing Figure 6, we state: “We have previously demonstrated that APT-1 regulates HRAS distribution between LR and DM microdomains in the plasma-membrane {Agudo-Ibanez, 2015 #667}. Since thyroid tumors behave differently depending on whether HRAS signals from DM or LR, it was of interest to understand how APT-1 levels related to tumor evolution”…., clearly explaining the rationale behind looking at APT-1 levels.

The reviewer has a good point suggesting that APT-1 expression could be regulated by RAS. To the best of our knowledge, there is no data in the literature in this respect. Looking at our unpublished data, we find no correlation between APT-1 expression levels and RAS (or BRAF) mutational status. However, the data that we have available comes from a small number of samples from different types of tumors, certainly far too preliminary to draw any conclusion. It is indeed a case worth investigating in further depth.

The results for VEGFB is also of interest, especially as many TKIs used in advanced thyroid cancer such as Lenvatinib are thought to mainly act through inhibition of VEGF which could be explored further in the discussion,

We are grateful to the reviewer for such suggestion. We have included the following sentence in the discussion: In this respect, it is worth indicating that lenvantinib and other Tyrosine Kinase Inhibitors, utilized for the treatment of advanced thyroid cancer, are known to inhibit VEGF receptors 1-3 { }, highlighting the importance of this signaling axis in thyroid neoplasia

do the authors have any protein expression results to support the differences in mRNA expression?

Yes, a blot showing VEGF-B protein levels has been incorporated in Figure 4B,C

In figure 6A the text labelling each group has been cut off and the figure is not of high enough resolution to see clearly, are the authors showing here that APTs had higher RNA expression from the TCGA study?

Yes, this figure shows APT1 expression levels from the TCGA database. The figure has been corrected following the reviewer’s suggestions. Since at a higher resolution it is impossible to make it fit, in the revised manuscript we present it, full size, as Supplemental Figure S2.

The authors should report the other common drivers such as BRAF and known prognostic mutations such as TERT to this figure

They have been included.

I cannot see how the authors can reach the conclusion that higher APT expression leads to improved survival, the most which could be said is that is it associated with a better overall survival but no P-value is reported, as nearly all the patients in the TCGA series had well differentiated PTCs it is likely many of the deaths reported are not due to thyroid cancer and for thyroid cancer deaths they are likely related to known drivers of aggressive disease, such as BRAF and TERT.

In Supp Fig S2 we have now included the p values for the correlations between APT-1 and the different driver mutations. We agree with the reviewer in that most that could be said is that high APT1levels  are associated with a better overall survival. We try to be conservative here, so we have changed the corresponding phrase to: In this respect, herein we unveil APT-1 as a potential indicator for favorable prognosis in thyroid papillary carcinomas. Clinical data obtained from TCGA shows that patients with high APT-1 levels are associated with better overall survival rates

Inhibitors of HRAS of tipifarnib are available as demonstrated in this paper https://cancerres.aacrjournals.org/content/canres/78/16/4642.full.pdf and should be discussed by the authors for further study of their model.

Good suggestion. At the beginning of the introduction we have included the following sentence: Drugs like the Farnesyl Transferase Inhibitor tipifarnib, effectively targeting HRAS, evoke beneficial cellular responses and extend survival in thyroid cancer animal models, pointing to this GTPase as a promising therapeutic target {Untch, 2018, }. 

In the conclusion the authors extend there results to NRAS but only one set of experiments use NRAS so I do not see how this can be supported.

Agree, we have somewhat overstated the NRAS case. We have therefore restricted the conclusions to HRAS.

Reviewer 2 Report

In this paper Garcia-Ibanez Y et all investigated the RAS subcellular localization in thyroid cancer cells. They found that the ability of tumors to extravasate and metastatize depends on RAS subcellular localization and that aggressiveness inversely correlated with tumor size.  They also show that this site-specific ability of RAS depends on VEGF-B secretion and that APT-1 alteration expression levels may regulate RAS subcellular  localization.

Minor points:

In figure 1A, explain why you should have a different PM of total, ER, and LR compared to DM and GC.

In figure 1 caption, at line 107, insert B).  

In Figure 2D, the statistical difference between LR and DM should be fixed with the dash more clear.

In Methods no mention about the authentication of cells is done. Please indicate if PCCL3, HTH83, and C643 have been authenticated and display the certificate.

Author Response

We thank the reviewer for his constructive suggestions

In figure 1A, explain why you should have a different PM of total, ER, and LR compared to DM and GC.

Agree, In the M&M section we have included the following sentence: Differences in MWs are due to the sizes of the tethering signals fused to either HRAS or CDC25. Specifically: M1, 7.2 kDa; LCK, 3.5 kDa; CD8, 45.2 kDa; KDELr, 44.4 KDa.

In figure 1 caption, at line 107, insert B).  

Corrected

In Figure 2D, the statistical difference between LR and DM should be fixed with the dash more clear.

Corrected

In Methods no mention about the authentication of cells is done. Please indicate if PCCL3, HTH83, and C643 have been authenticated and display the certificate.

Agree, we have now included the following statement: All cell lines used in this work have been authenticated every 12 months by short tandem repeat profiles using the Applied Biosystems Identifier kit in the Genomic Facility at the Instituto de Biomedicina y Biotecnología de Cantabria (IBBTEC, Spain).

Reviewer 3 Report

Comments to the Authors:

This manuscript is attractive focusing on the relationship between RAS subcellular localization and biological behavior and agressiveness of thyroid tumors. The role of VEGF-B and APT-1 in RAS-driven tumor growth and dissemination have been also established. This work is largely original and well-conducted with the interesting conclusions having the potential clinical effects.The manuscript is well written. However, several points should be addressed:

Section 1 (Introduction):

  • clarify the different plasma membrane compartments: lipid rafts and disordered membrane
  • consider to put Figure of RAS location within the cell
  • try to specify more the RAS signaling pathways and effectors and consider to put the corresponding figure

Section 3 (Discussion)

Make clear the fact that you looked for APT-1 alterations in databases of thyroid tumors only in a cohort of papillary thyroid carcinomas, while follicular thyroid carcinomas or anaplastic carcinomas harbor more RAS mutations.

Section 4 (Materials and Methods):

Specify more cell cultures:

PCCl3 … well differentiated rat thyroid follicular cell line (wild type cells)

C643 and HTH83… oncogenic H-RAS mutated human anaplastic thyroid cell lines 

Minor comments:

Line 24: VEGF-B should be defined

Line 25: change …acyl-thioesterase APT-1 to Acyl Protein Thioesterase 1 APT-1 (originally designated lysophospholipase I; LYPLA1)

Line 99: link to „Fig. 1B“ ……. However, there is no the letter „B“ indicated in Figure 1 (lines 103-111)

Author Response

We thank the reviewer for his positive comments

Make clear the fact that you looked for APT-1 alterations in databases of thyroid tumors only in a cohort of papillary thyroid carcinomas, while follicular thyroid carcinomas or anaplastic carcinomas harbor more RAS mutations.

In the Results section, we describe in further detail the TCGA cohort that we have utilized:  “…we selected a cohort of 507 samples of thyroid tumors, of which 77.3% correspond to papillary carcinomas; 20.7% follicular carcinomas; and 2% to poorly differentiated tumours”…… We have also indicated likewise in the discussion: … Clinical data obtained from TCGA, mostly corresponding to papillary thyroid carcinomas, shows that patients with high APT-1 levels are associated with better overall survival rates.

Section 4 (Materials and Methods):

Specify more cell cultures:

PCCl3 … well differentiated rat thyroid follicular cell line (wild type cells)

Corrected

C643 and HTH83… oncogenic H-RAS mutated human anaplastic thyroid cell lines 

Corrected

Minor comments:

Line 24: VEGF-B should be defined

Defined in line 89

Line 25: change …acyl-thioesterase APT-1 to Acyl Protein Thioesterase 1 APT-1 (originally designated lysophospholipase I; LYPLA1)

Corrected in line 28

Line 99: link to „Fig. 1B“ ……. However, there is no the letter „B“ indicated in Figure 1 (lines 103-111)

Corrected

Reviewer 4 Report

Io

The paper presented by Yaiza García-Ibáñez, Garcilaso Riesco-Izaguirre, Pilar Santisteban, Berta Casar and Piero Crespo, reported an interested and innovative work about RAS genes.

The authors observed a thyroid tumor different behavior associated with the sub-localization of oncogene RAS in the cells.

In the paper, the authors presented well performed in vitro and in vivo experiments, for this reason I suggest only few minor revisions:

  • I suggest explaining, even only at a theoretical level, the double bands presented in western blots (e.g. Fig 1A).

  • Since the paper showed is one of the first observations of a phenomenon that should be verified in vivo in a cohort of patients, I suggest to mitigate the language underling that in this work there are in vitro experiments that indicate an association between RAS genes subcellular localization and tumor behavior.

Author Response

We thank the reviewer for his positive comments

I suggest explaining, even only at a theoretical level, the double bands presented in western blots (e.g. Fig 1A).

If the reviewer refers to the band of lower MW that appears in Fig 1A lane “DM”, corresponding to the construct HA-HRAS-CD8, we believe that it must be some truncated form expressing HA-HRAS and a shorter version of the CD8 alpha receptor. We have found that, irrespective of the cell type, sometimes it appears, sometimes it doesn’t. Though we have never observed any differences with respect to HRAS activity depending on it

Since the paper showed is one of the first observations of a phenomenon that should be verified in vivo in a cohort of patients, I suggest to mitigate the language underling that in this work there are in vitro experiments that indicate an association between RAS genes subcellular localization and tumor behavior.

Agree. In the abstract we now mention that the study refers to the chick embryo metastasis model, and in the discussion we include the phrase: “ Studies in a broader panel of clinical cases will be necessary in order to reach a final conclusion in this respect”.

Round 2

Reviewer 1 Report

The authors have responded appropriately to the questions and improved the manuscript which adds important information to the biology of HRAS driven thyroid cancer.